# A novel rocket borne ion mass spectrometer with large mass range: instrument description and first flight results

**Joan Stude[2], Heinfried Aufmhoff[2], Hans Schlager[2], Markus Rapp[2,1], Frank Arnold[4,2], and Boris Strelnikov[3]**

[1]Ludwig-Maximilians-Universität München (LMU), Atmospheric Physics, München, Germany
[2]German Aerospace Center (DLR), Institute of Atmospheric Physics, Oberpfaffenhofen, Germany
[3]Leibniz-Institute of Atmospheric Physics (IAP), Kühlungsborn, Germany
[4]Max-Planck-Institute for Nuclear Physics (MPIK), Heidelberg, Germany

**Correspondence:** Joan Stude (joan.stude@dlr.de)

**Abstract.** We present a novel rocket borne ion mass spectrometer ROMARA (ROcket borne MAss spectrometer for Research in the Atmosphere) for measurements of atmospheric positive and negative ions (atomic, molecular and cluster ions) and positively and negatively charged meteor smoke particles. Our ROMARA instrument has, compared to previous rocket borne ion mass spectrometers, a markedly larger mass range of up to $m/z$ 2000 and a larger sensitivity, particularly for meteor smoke particle detection. Major objectives of this first ROMARA flight included: a functional test of the ROMARA instrument, measurements between 55 km and 121 km in the mass range of atmospheric positive and negative ions, a first attempt to conduct mass spectrometric measurements in the mass range of meteor smoke particles with mass to charge ratios up to $m/z$ 2000, and measurements inside a polar mesospheric winter echo layer as detected by ground based radar. Our ROMARA measurements took place on the Arctic island of Andøya/Norway around noon in April 2018 and represented an integral part of the PMWE rocket campaign. During the rocket flight, ROMARA was operated in a measurement mode, offering maximum sensitivity and the ability to qualitatively detect total ion signatures even beyond its mass resolving mass range. On this first ROMARA flight we were able to meet all of our objectives. We detected atmospheric species including positive atomic, molecular and cluster ions along with negative molecular ions up to about $m/z$ 100. Above $m/z$ 2000, ROMARA measured strong negative ion signatures, which are likely due to negatively charged meteor smoke particles.

## 1 Introduction

Meteor smoke particles (MSPs) are of considerable current interest, since they have several interesting atmospheric roles: MSPs may act as sites of heterogeneous reactions involving atmospheric trace gases and ions. Moreover, MSPs act as nuclei in the formation of mesospheric water ice clouds(Rapp and Thomas, 2006) and as nuclei in the formation of the stratospheric sulphuric acid (Arnold and Fabian, 1980; Arnold et al., 1981; Hervig et al., 2017) and nitric acid aerosol layers (Arnold and Knop, 1989; Arnold et al., 1989; Voigt et al., 2005; Frankland et al., 2015; James et al., 2018; Curtius et al., 2005), which have an impact on ozone and climate (Crutzen and Arnold, 1986). MSPs may also influence the charge balance of the lower ionosphere, by acting as scavengers of free electrons and ions (Schulte and Arnold, 1992; Rapp and Lübken, 2001; Friedrich et al., 2012). Furthermore, electrically charged MSPs have been proposed to play a potential role in the formation of polar mesospheric winter radar echoes (PMWE) (Rapp et al., 2011; La Hoz and Havnes, 2008).

MSPs are formed by the ablation of meteors or interplanetary dust particles in the upper atmosphere, leading to meteoric vapours, which ultimately recondense to secondary aerosol particles, as was originally hypothesized by Rosinski and Snow (1961). The meteoric vapours are released, mostly at altitudes around 90 km in the mesopause region, during the entry of the atmosphere at high velocities (Kalashnikova et al., 2000; Plane, 2003). Hereafter, such vapours undergo gas phase reactions with atmospheric gases and ions and ultimately recondense, leading to tiny aerosol particles (Plane, 2012). Hunten et al. (1980) termed these particles meteor smoke particles and conducted model simulations, predicting

a MSP layer to be present between 70 and at least 100 km, peaking around 85 km. Predicted MSP radii range from 0.2 to 10 nm, corresponding to hundreds to millions of atomic mass units (u) (Fig. 1). For example, at 85 km, MSPs with radii larger than 0.2 nm have a predicted number concentration of $7 \cdot 10^4$ cm$^{-3}$. The meteor vapours, which lead to MSP formation, are to a large part composed of metal and silicon atoms, formed mostly via ion-molecule reactions with atmospheric positive ions. Early rocket borne ion composition measurements of the whole suite of the most abundant meteoric positive ion species Fe, Mg, Si and their isotopes revealed an ion composition similar to the elemental composition of chondrites (Krankowsky et al., 1972). The mass density of ordinary chondrites is about 3.5 g cm$^{-3}$ (Britt and Consolmagno, 2003), however the ablation process might form particles of lower densities with 2.0 g cm$^{-3}$ as assumed by Hunten et al. (1980) and Plane et al. (2014).

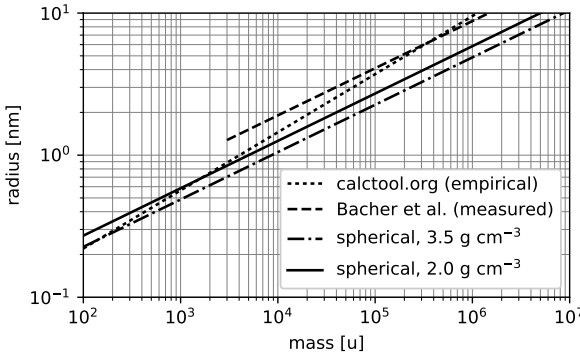

**Figure 1.** Comparison of different relationships between molecule mass and size with data from Bacher et al. (2001) and (Shipway, A. and Shipway, S., 2008).

Using the above values in Fig. 1 and assuming a spherical MSP particle of 0.3 nm radius, one would estimate an effective mass of about 135 u to 240 u. First observational indications for the presence of such negatively charged nascent MSPs (>473 u) have been obtained from rocket borne measurements of ion mass spectrometers by Schulte and Arnold (1992) under twilight conditions. Meanwhile several rocket borne electrostatic probe measurements provided evidence for larger MSPs than measured by the ion mass spectrometers (Gelinas et al., 1998; Rapp et al., 2005; Lynch et al., 2005; Strelnikov et al., 2012; Robertson et al., 2014; Havnes et al., 2015; Asmus et al., 2017). Even some first direct information on the chemical nature of MSPs was obtained by rocket borne photo ionisation measurements. It was found that the ionisation potential of MSPs is somewhat similar to that of Fe and Mg hydroxide clusters (Rapp et al., 2012). Besides those direct in-situ measurements, more indirect measurements of MSP-signatures in the spectra of incoherent scatter radars were reported by several investigators: Rapp et al. (2007); Strelnikova et al. (2007); Fentzke et al. (2009,

2012). Also satellite measurements with the Solar Occultation For Ice Experiment (SOFIE) have provided clear evidence of MSP in solar occultation extinction measurements (Hervig et al., 2009) .

The present paper reports on a search of MSPs using a novel rocket borne ion mass spectrometer, having an increased mass range of up to $m/z$ 2000 and an increased sensitivity. Importantly, the rocket flight was determined to penetrate a PMWE layer, which could indeed be realized. Details about the PMWE campaign as such will be given in Strelnikov et al. (2020) and Staszak et al. (2020). Here we give a thorough description of the ROMARA instrument and a brief presentation of ion and MSP data.

## 2   Instrument description

The instrument ROMARA is a cryogenically pumped quadrupole mass spectrometer based on earlier designs at Max-Planck-Institute for Nuclear Physics (MPIK) (Arnold et al., 1971; Krankowsky et al., 1972; Arnold et al., 1977b). A cross-section of instrument and mass spectrometer is shown in Fig. 2. Ions enter the instrument through a knife edge intake orifice with a radius of 0.5 mm on the tip of a double cone. The design allows to sample the atmosphere in front of the shock due to the supersonic speed of the payload. A quadrupole lens (Brubaker, 1968) of 36 mm length is mounted between intake orifice and quadrupole, with skewed tips for a placement as close as possible to the intake cone. The quadrupole mass filter has a length of 115 mm and uses cylindrical rods of 2.4 mm radius. The field radius $r_0$ between the rods is given by a ratio of $r_{rod}/r_0 = 1.128$ as recommended by Douglas (2009).

Ions passing the mass filter are detected by a channel electron multiplier (CEM), placed in the centre of the instrument, in line of sight of the intake orifice. Quadrupole and detector are almost completely surrounded by the cryopump which serves also as mechanical support. A cap seals the instrument intake during storage and launch and is jettisoned before the measurements begin. For this purpose a spring loaded bayonet ring is turned by pyro-actuators to release the cap. Inside the cap a commercial ion source (electron ionisation, Hiden Analytical 205011) allows to calibrate and test the instrument on the ground and monitors operation during launch until the cap together with the ion source is removed. To increase ion transmission, independent bias potentials can be applied to the intake cone and the quadrupole lens and rods (see 2.4). To maximize detection efficiency, especially for heavy ions, the CEM (Photonis 4830-MgO) is coated with magnesium oxide and biased to ±1800 V with a constant voltage across the CEM of about +2700 V. Measuring positive ions requires all bias voltages to be negative and vice versa. For an alternating operation of the instrument between positive and negative ions, the bias voltages are switched in the dead time between the spectra.

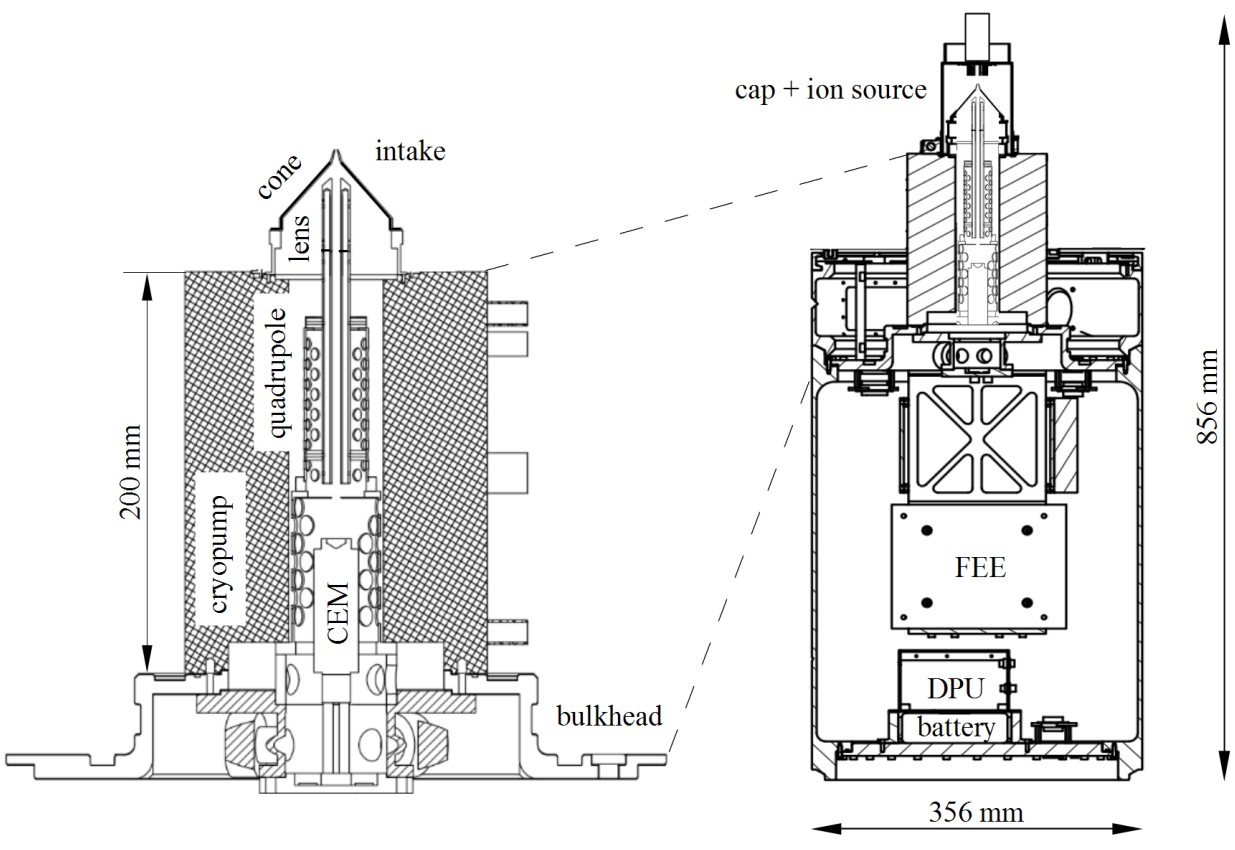

**Figure 2.** LEFT: detailed cross-section of ROMARA mass spectrometer in flight configuration. RIGHT: cross-section of complete ROMARA instrument in launch configuration. CEM: channel electron multiplier, FEE: front-end electronics, DPU: data processing unit.

The cryopump is a bath type design using gold plated cooling surfaces. A heat shield, cooled by the evaporating cryogen is placed between reservoir and outer shell. As cryogen, liquid helium is most readily available, however liquid helium evaporates rather quickly and the reservoir would be depleted in about an hour. Longer standby times can be achieved using neon because of its higher vaporisation enthalpy. Thus the rocket can be safely prepared for launch and the probability to meet the desired atmospheric conditions during a day of work is higher. On the day of launch a total standby time of $\approx 7\ h$ with subsequent successful operation was achieved. We produced liquid neon at the launch site through liquefying neon gas with liquid helium. A disadvantage of using neon as cryogen is the inefficient pumping of hydrogen, helium and neon itself due to their boiling points below that of neon. While parts of these gases are trapped in the frozen air building up on the cooling surfaces, the remaining gas (mostly helium) could accumulate above a critical pressure. However, for our combination of intake size and duration of flight this is not of concern.

A reinforced bulkhead holds the complete construction of electronics, quadrupole and cryopump. The bulkhead further seals the main structure against atmosphere and water during splashdown. Front-end electronics (FEE), data processing unit (DPU) and batteries are housed in the main structure at 1.5 bar absolute nitrogen pressure to safely handle high voltages. The whole instrument is 856 mm in length with a diameter of 356 mm at a total mass of about 50 kg.

## 2.1 Principle of operation

A quadrupole mass spectrometer separates ions by their mass-to-charge ratio ($m/z$) in applying electrical fields along a central drift path of ions. These fields are ideally hyperbolic but are often approximated by cylindrical electrodes, the rods, circularly arranged along their length. The required fields are formed by radio frequency ($V_{RF}$) and static ($V_{DC}$)

potentials with opposite polarity at neighbouring rods. Thus opposing rods are electrically connected and form two pairs of rods. Ions of a certain $m/z$ retain stable trajectories, pass the quadrupole and can be detected at the exit, while other ions collide with the rods and are lost. To pull ions in between the rods a constant bias voltage $V_B$ is applied to all rods, i.e.,

$$V_{rod} = V_B \pm [V_{DC} + V_{RF}\cos(\omega t)] \qquad (1)$$

The quadrupole allows 2 modes of operation: an RF-only mode and a line mode. In RF-only mode, $V_{DC}$ is set to zero. With increasing $V_{RF}$ lower masses are rejected, until a maximum $V_{RF}$ is reached. The count rate in the detector will thus eventually drop for each ion mass and produce a step in the recorded mass spectrum, which can be analysed for width and height. Ions with masses above the mass given by the maximum $V_{RF}$ still pass the quadrupole but can not be mass analysed. This is often described as a high pass mass filter, although it is actually a wide bandpass as illustrated in Fig. 3. We simulated transmission curves for different mass settings of the quadrupole and different ion masses using SIMION® (Dahl, 2000) to model the electrical fields and individual ion trajectories. Collisions with the background gas or charge exchange processes were omitted. For each ion mass per charge and mass per charge setting of the quadrupole, a population of 3000 ions is started inside the intake orifice with a constant velocity of $980\ \mathrm{m\,s^{-1}}$ (= rocket velocity, $v$), an angle of attack of $2.2°$ and a uniform conical direction distribution with an opening angle $\delta$ defined as:

$$\delta = 2\ \mathrm{atan}\left(\frac{3kT}{mv}\right) \qquad (2)$$

with $k$ as the Boltzmann constant, $T$ as temperature (180 K), $m$ as ion mass.

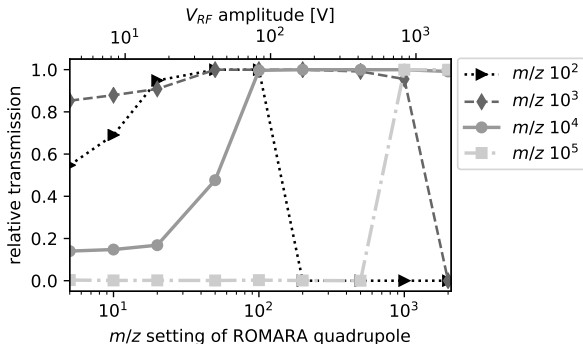

**Figure 3.** SIMION® simulation of ROMARA quadrupole for heavy ion transmission in RF-only mode for an angle of attack of $2.2°$.

It is evident that for low $V_{RF}$ values and hence mass settings of the quadrupole, only small fractions of ions above $10^4$ u ($\approx 1.2$ nm radius) pass the quadrupole to the detector.

Ions with sufficient masses and thereby sufficient kinetic energy do not respond efficiently to the small applied fields. In this case the intake orifice and exit aperture act as collimator and limit the detector flux to trajectories smaller than $0.85°$ from the central axis. The instrument thus mass analyses particles up to $m/z$ 2000 and detects the presence of particles up to about $m/z$ $3{\cdot}10^5$ ($\approx 3$ nm radius). For the line mode with an additionally applied $V_{DC}$, this effect is much smaller and can be neglected. For the quadrupole to operate in line mode $V_{DC}$ is set to a constant ratio to $V_{RF}$ and thus forms a narrow band pass mass filter. With increasing voltages the band pass window is moved from low to high masses and a line spectrum can be recorded. The ratio of $V_{DC}$ to $V_{RF}$ determines the size of the band pass window and thus resolution and sensitivity. The voltage applied to the quadrupole lens always corresponds to $V_{RF}$ only. This minimizes ion losses at the entrance to the rod system, when the quadrupole is operated in line mode.

## 2.2 Ion sampling from the atmosphere

The payload moves at supersonic speeds through the atmosphere, developing a shock in the ram direction. The knife-edge double cone is designed to sample ions in front of the shock, avoiding perturbations and possible brake up of weakly bound ions. The shock was simulated with the direct Monte Carlo simulation software DS2V (Bird, 1988, 1994) under conditions of a standard atmosphere with a composition of nitrogen and oxygen (NRLMSISE-00) and rocket speeds appropriate for our flight. The left panel of Figure 4 shows rocket speed, temperature and number density for different altitudes used in the simulation. The right panel of Figure 4 shows the relative air speed and the increase of temperature and number density 1 cm upstream the intake orifice. Up to 80 km the ratios stay about unity. Above 80 km the shock starts to detach from the payload and number density and temperature begin to increase, while the relative air speed decreases. Values change roughly by a factor of 2 at 100 km altitude as compared to undisturbed conditions. Particles with much larger mass such as MSPs, will be less affected by the shock formed by air molecules (Hedin et al., 2007; Asmus et al., 2017) and thus penetrate it more efficiently.

## 2.3 Electronics

As front-end electronics we used modified electronics from MPIK, originally developed for aircraft operations e.g.: the quadrupole power supply. The electronic pulses from the CEM are transformed by an Amptek A111F charge amplifier to digitally countable pulses. To control the front-end electronics and process the data we use a microprocessor/FPGA system from National Instruments (sbRIO-9637). During ground preparation the system can be remotely controlled and updated if necessary. After launch the system is

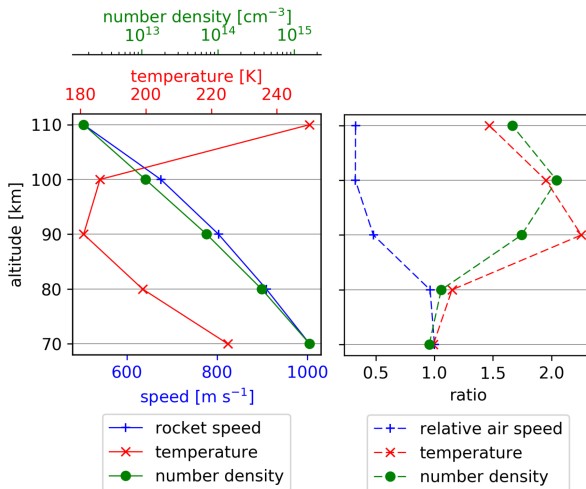

**Figure 4.** LEFT: DS2V simulation input parameters from NRLMSISE-00 Atmosphere Model, RIGHT: DS2V simulation results, 1 cm upstream of inlet orifice, plotted as ratio to ambient values of relative air speed, atmospheric temperature and number density.

controlled by a time-line program that does not allow further interactions. The data are transmitted via the rocket service module but are also stored on an internal SD card. The data volume is about 10 kbyte per spectrum. The instrument is self powered using lithium iron phosphate batteries which allow approximately 1.5 h of operation. A mass spectrum is divided into 4096 mass channels with a dwell time of 300 $\mu$s. This results in a total time of 1.274 s, including 0.045 s of dead time at the beginning of each spectrum. The total time of a spectrum and thus height resolution during flight is mostly limited by the $V_{RF}$ oscillator. The oscillator runs at about 1.4 MHz with a maximum amplitude ($V_{RF}$) of $\approx$ 1750 V.

## 2.4 Calibration

The bias voltages of the quadrupole lens and the rods were calibrated using positive xenon ions. The intake cone potential was set to zero and therefore payload ground. We found an optimum transmission if the rods have a potential of -50 V and the quadrupole lens -20.5 V. For negative ions the same absolute values were used. During tests we observed a significant loss of transmission if both voltages were equal. For the mass calibration of the instrument we used 4 eV ions of neon, krypton, xenon and perfluorotributylamine (PFTBA, Heptacosa, FC-43), allowing a calibration up to $m/z$ 502. In Fig. 5 we show in the upper panel measurements of Kr and Xe ions, the cumulative distribution function (CDF) fits of the ion mass steps, the reconstructed Gaussian peaks and the respective isotopes lines from the National Institute of Standards and Technology (NIST). The mass resolution $m/\Delta m$ at 50% peak height was determined with about 17.5 ($\approx$ 5 u peak width for Kr) and the theoretical mass range from $m/z$ 5

to $m/z$ 2075. In the lower panel we show the standard spectrum of PFTBA from NIST data and the measured spectrum of ROMARA. The NIST spectrum was converted to an RF-only version by summing up all single peaks in the NIST data. The measured ROMARA spectrum was normalized to the $m/z$ 69 peak and shows good agreement for the major steps of PFTBA up to $m/z$ 502. Steeper falloffs between the major PFTBA steps are a result of typical hydrocarbon contaminations of the ion source (steps $< m/z$ 300, Fig. 6A). In cases where there are many small peaks close to each other the ROMARA spectrum turns into a continuous slope because of the limited mass resolution.

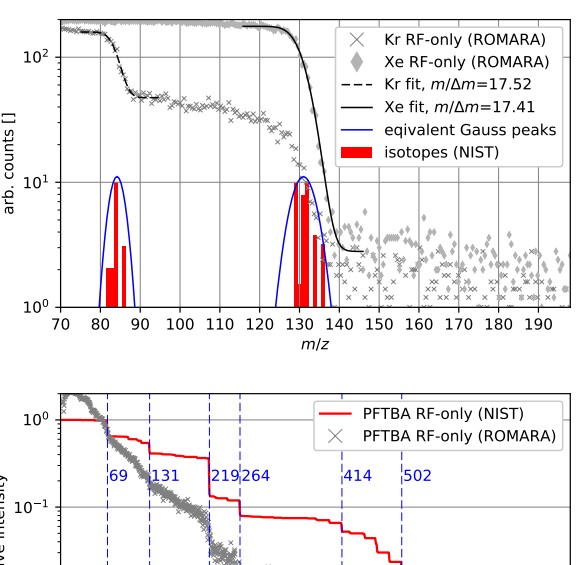

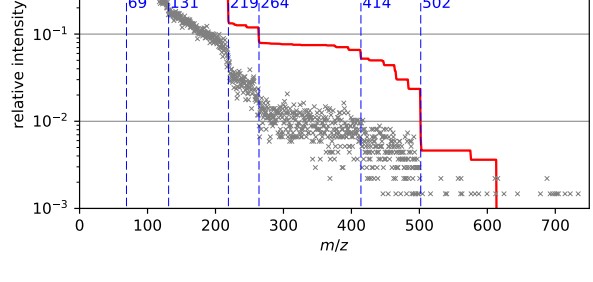

**Figure 5.** UPPER panel: RF-only spectrum of Kr and Xe, CDF fit and equivalent Gauss peak, LOWER panel: ROMARA RF-only measurement of PFTBA and NIST data of PFTBA

The noise level of the instrument is mostly determined by the oscillator and the switching between positive and negative ion mode leading to increased counts at the beginning and at the end of each spectrum, especially in the negative ion mode. On the launch pad with the instrument operating nominally and the ion source switched off, the average noise floor over a whole spectrum was 9 Hz in positive ion mode and 240 Hz in negative ion mode, well below our 1 count limit of 3.3 kHz.

## 3 Measurement and discussion

### 3.1 Launch conditions

The main launch criteria were determined by the MAARSY radar (Latteck et al., 2012; Rapp et al., 2011; Latteck et al., 2019), looking for polar mesospheric winter echoes. The radar pointed alternately along the rocket trajectory and to zenith (see Latteck et al. (2019) for details). The days before launch several echoes were detected but other launch criteria for e.g. the sea recovery were not met. On 13 April 2018 at 09:44:00 UTC, ROMARA was launched directly into an echo between 78 km and 80 km altitude, with weaker echoes visible just before at around 65 km and 70 km. After separating the nose cone at 52.2 km the instrument cap was jettisoned at 54.5 km and ROMARA measurements began. At 60.0 km the motor was separated from the payload. The payload hit the echo at its decaying tail before reaching an apogee of 121.4 km. The electron density was measured with the on-board wave propagation experiment (Friedrich et al., 2013) but also with the Saura MF radar (Latteck et al., 2019) about 20 km south of the launch site. The ionosphere was moderately disturbed with a simulated riometer absorption of 0.26 dB at 27.6 MHz. At the given time and location a solar zenith angle of 61.6° is calculated, the direction of the launch was 330° azimuth (Northwest by north).

### 3.2 Ion measurements

The instrument was operated in RF-only mode throughout the flight and measured natural ions during ascent from 54.5 km to apogee at 121.4 km but also on the downleg in the rocket wake. The intake cone was applied with a constant 0 V bias and thus was at payload potential during the whole flight. In this instrumentally oriented paper, we present 8 exemplary mass spectra, obtained during rocket ascent: 4 positive and 4 negative. These include 2 test spectra, obtained just prior to cap-ejection and 6 ambient ion spectra. A more detailed analysis of the ion measurements will be presented elsewhere. The internal ion source was operated up to 49 km altitude and was switched off before the ejection of the cap. Other spectra were chosen around 70 km and 106 km. At 70 km the sampling conditions are ideal with the shock wave being well attached to the intake cone as the simulations of Fig. 4 show. Thus ions enter the instrument with minimum disturbance and under small angles of attack ($\alpha \approx 2.2°$). In contrast to 70 km, the spectrum at 106 km is much more affected by the shock, now completely detached and angles of attack are considerably larger ($\alpha \approx 11.4°$).

Raw count rates $c_{raw}$ were corrected for detector dead time $\tau$ and angle of attack $\alpha$. The angle was provided by the rocket operator and the dead time was taken from the A111F data sheet with $\tau = 350$ ns, giving the corrected count rate $c$:

$$c = \frac{c_{raw}}{(1 - c_{raw}\tau) \cdot \cos(\alpha)}. \tag{3}$$

In Fig. 6 and Fig. 7 we included an 8 channel mean, as the original data is noisy. The count rates given hereafter refer to the 8 channel running mean count rate.

### 3.3 Positive ions

Figure 6A depicts a positive test spectrum obtained at 46.8 km, prior to cap ejection with the internal ion source in operation. In contrast to the calibration data of Fig. 5 where a laboratory power supply for the ion source is used, the internal power supply provides a less stable current and typical residual gas steps are not clearly visible. However the count rate clearly decreases from about $m/z$ 28 up to $m/z$ 300, typical for leaking air and contaminations in the ion source and cap, e.g. hydrocarbons. The maximum count rate is 636 kHz for below $m/z$ 28.

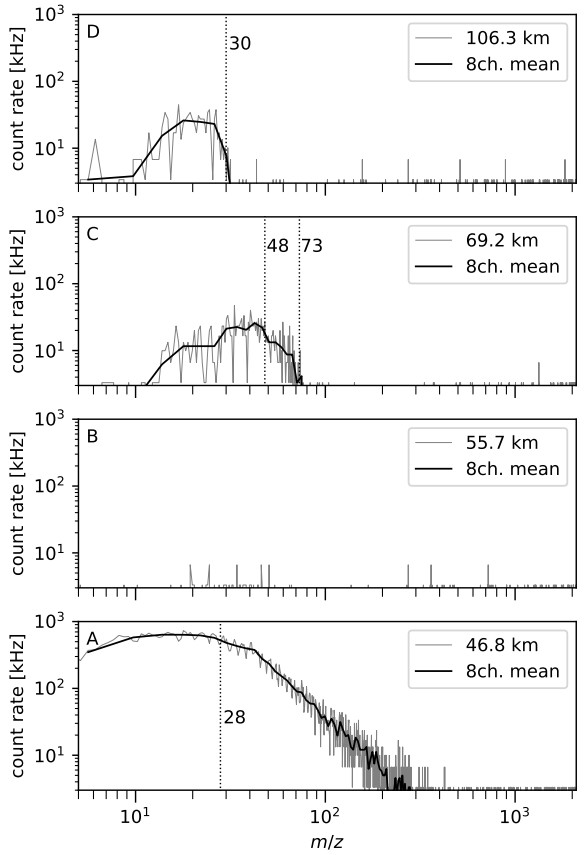

**Figure 6.** Positive ion mass spectra at 4 different altitudes during ascent in logarithmic scales. Spectrum A shows the residual gas spectrum from the internal ion source before ejecting the cap. Spectrum B, C and D show natural ambient ions. The vertical resolution for spectrum B, C and D is 1426 m, 1270 m and 678 m.

Figure 6B depicts the spectrum at 55.7 km, just after ejecting the cap, and serves to show that no residuals are present in the mass spectrum. Some counts of the same type as in the spectrum of Fig. 6C are already visible.

Figure 6C depicts a spectrum starting at 69.2 km altitude with ambient atmospheric positive ions. It has a maximum mean count rate of about 26 kHz and a maximum of $m/z$ 76, which is most likely due to the expected proton hydrate $H^+(H_2O)_4$[73 u]. This proton hydrate of 4th order has been measured previously (Arnold et al., 1977a; Kopp et al., 1984) often together with $H^+(H_2O)_3$[55 u] and higher orders. However, in this particular spectrum a step is found at around $m/z$ 49 which is thus more likely due to $NO^+(H_2O)$[48 u]. A less defined step around $m/z$ 58 with only few counts per mass channel, corresponds likely to $H^+(H_2O)_3$[55 u].

In Fig. 6D a spectrum at 106.3 km is shown with a similar maximum count rate of 26 kHz with a step around $m/z$ 28. This is consistent with $NO^+$[30 u] or $O_2^+$[32 u] as the most dominant ions at that altitude.

It is noticeable that the spectra do not start at maximum count rates. This is qualitatively the same effect as shown by the 100 u curve in Fig. 3.

None of these positive ion spectra indicate the presence of heavy positive ions.

### 3.4 Negative ions

In Fig. 7 we show characteristic negative ion spectra for similar altitudes as in Fig. 6. The spectrum in Fig. 7A is taken before cap ejection at an altitude of 48.3 km with the ion source operating. As expected, negative ions are not present, as they are not formed by the ion source (electron ionisation) in the cap.

Figure 7B shows the spectrum beginning at 54.2 km altitude, in which the cap was ejected. Before cap ejection the count rate is at the same low noise level as in Figure 7A. Instantaneously with cap ejection around $m/z$ 400, the count rate increases to a plateau around $m/z$ 560 with a maximum count rate at 200 kHz. Hereafter the count rate decreases to about 50 kHz showing some minima and maxima, possibly related to rocket spin To the end of the spectrum the count rate remains at about 50 kHz indicating ions beyond our mass range of $m/z$ 2000.

Figure 7C at 70.5 km altitude contains light negative ions with a maximum count rate of 200 kHz exceeding the positive count rate almost tenfold. The count rate in the end of the spectrum is about 100 kHz. The inset is a blow up of the small ion mass range to $m/z$ 100, indicating the presence of numerous unresolved mass steps with high count rates even at the smallest mass channels. Potential mass steps can be found around: $m/z$ 24, $m/z$ 48, $m/z$ 58, $m/z$ 65 and $m/z$ 79. These steps could be caused by $CN^-$[26 u], $Cl^-(H_2O)$[53 u], $CO_3^-$[60 u], $HCO_3^-$[61 u], $NO_3^-$[62 u], $CO_4^-$[76 u], of which some have been measured previously by Arnold et al. (1971, 1982) and Kopp (1992). The heavy ion signature is now clearly modulated by the rocket spin (3.6 Hz) with the maxima being about 0.27 s apart and thus requiring 4.5 modulations of the incident ion flux within a

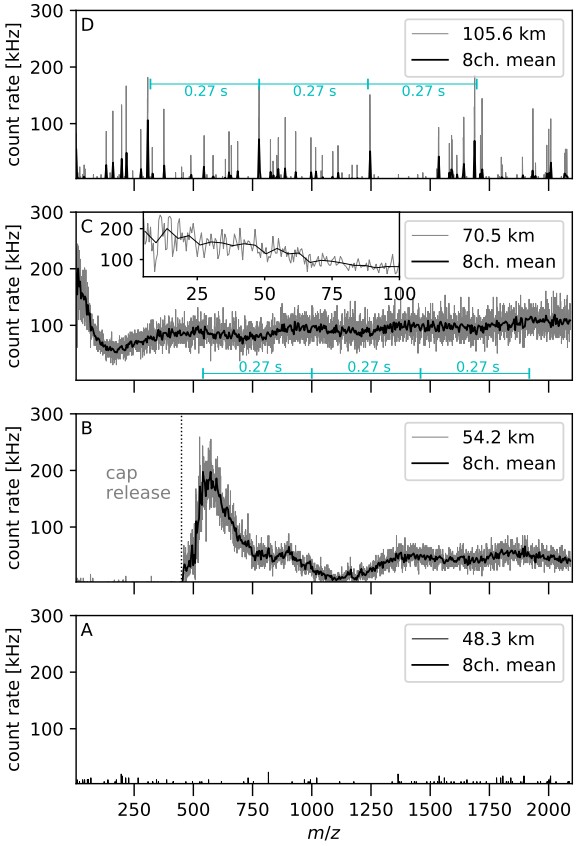

**Figure 7.** Negative ion mass spectra in linear scales during ascent, at similar altitudes as the positive ion mass spectra. In spectrum A the instrument is still sealed to the environment, while in spectra B, C and D natural ambient ions are present. The vertical resolution for spectrum B, C and D is 1431 m, 1245 m and 688 m.

full spectrum. This can only be caused by asymmetries in the ion optics or at the intake orifice under angles of attack larger than zero. We therefore interpret this signal as negative ions with masses above our mass range of $m/z$ 2000. Other origins of this signal such as stray UV-light, neutral particles or a pressure effect inside the instrument can most likely be ruled out: while UV-light or neutral particles could trigger false counts, this would also be visible in the positive ion mode as well and was tested for UV-light in the laboratory. A pressure effect inside the instrument due to the higher voltages of the negative ion mode would lead to counts that directly correlate to decreasing ambient pressure but instead we measured an increase from about 50 kHz in Fig. 7B to about 100 kHz in Fig. 7C.

In Fig. 7D at 105.6 km altitude, increased levels of noise can be seen with no obvious small or large ion signatures. However, the largest spikes at: $m/z$ 1688, $m/z$ 1245, $m/z$ 778 and $m/z$ 308 are 0.27 s apart and thus indicate some heavy ions, again modulated by the rocket spin.

## 3.5  Charge balance

It is usually assumed that the atmosphere is quasineutral, such that the positive ion density, balances the negative ion and electron density: $N^+ = N^- + N_e$. At around 70 km we measured a maximum count rate of 26 kHz for positive ions and around 300 kHz for negative ions (light ions + heavy ions). Count rates may be converted to ion densities by the air column swept out through the intake orifice and rocket velocity: $N = c/(\pi r^2 v)$. This leads to ion densities of $N^+$ = 32 cm$^{-3}$ and $N^-$ = 379 cm$^{-3}$. In Latteck et al. (2019) the electron density at 70 km was about 500 cm$^{-3}$, suggesting about 27 times more negative charges than positive. At around 55 km the situation is even worse with very few positive counts. The simple conversion of counts to ion density however does not consider the transmission efficiency of ions between intake and CEM, the absolute detection efficiency of the CEM and the payload potential as the most obvious factors which could lead to unequal or larger ion densities. The transmission efficiency of the quadrupole in RF-only mode is assumed to be close to unity (Schulte and Arnold, 1992) and should be equal for both charge states as the potential amplitudes are identical between positive and negative ion mode. The absolute detection efficiency across the mass range of our instrument, for the MgO coated CEM is presently unknown. For the same CEM model but without MgO coating and similar ion energies, Keller and Cooper (1996) found the absolute detection efficiency for oxygen to be about 0.6 for positive ions and 0.7 for negative ions. Heavier atomic ions generally have a lower detection efficiency (Krems et al., 2005), e.g. xenon: 0.15 at 2 keV but publications are scarce for heavier molecules. Measurements from Gilmore and Seah (2000) on micro channel plates, indicate a rather low detection efficiency of around 0.02 for molecules of 2352 u at 2 keV. A positive payload potential and therefore a positive potential on our intake cone could decrease positive ion count rates and increase negative ion count rates. Such a potential is the result of currents to and from the payload interacting with the surrounding plasma, mainly dominated by free electrons (Darian et al., 2017; Friedrich et al., 2013; Sternovsky et al., 2004). At 70 km and below the electron density is low and the presence of heavy negative ions as carriers of negative charges could eventually lead to a positive payload potential as they are much slower than free electrons and the dominating positive ions. However, other effects such as photoeemmision or triboelctric charging (Havnes and Næsheim, 2007; Antonsen et al., 2019) could lead to positive charging. In the D-region, payload charging has been known as a major error source for quantitative rocket borne measurements for a long time (Thrane, 1986). Hence, the effect of payload charging on the here presented observations with its apparent departure from charge neutrality needs to be investigated in more detail. A full treatment of payload charging effects is beyond the scope of this investigation and will be dealt with in detail in a future publication.

## 4  Conclusions

With our rocket borne mass spectrometer for research in the atmosphere (ROMARA), we successfully re-vitalized an instrument concept to make mass spectrometric measurements in the mesosphere and lower thermosphere. With its extended mass range for MSPs, ROMARA detects ambient atmospheric positive and negative ions up to $m/z$ 2000 and in addition, ROMARA measures the total count rate of ions with masses above $m/z$ 2000. We have simulated the instruments aerodynamic and ion-optical behaviour and conducted laboratory measurements for the characterisation of ROMARA. The first ROMARA flight, which took place at noon on April 13th 2018 and reached an apogee of 121.4 km, was successful in detecting ambient atmospheric positive and large negative ions. After the flight the instrument was recovered nearly undamaged. Six exemplary mass spectra of ambient atmospheric positive and negative ions, measured during rocket ascent, are shown in the present paper and demonstrating the successful application of an instrument to conditions in the middle atmosphere.

The most important scientific results of the ROMARA data presented here concerns the detection of large negative ions and a strong indication for the absence of large positive ions. Most likely, the large ions are actually negatively charged meteor smoke particles with radii of about 0.6 nm to 2.5 nm ($m/z$ 2000 - $m/z$ 10$^5$). Besides large negative ions also small negative ions with masses mostly below $m/z$ 150 have been detected at 70 km. At 106 km, no negative ions have been detected. These findings are consistent with previous measurements. Positive ions measured at 70 km are mostly hydrated cluster ions up to $m/z$ 73. At 106 km the detected small positive ions have mass numbers around $m/z$ 30. Again, this is consistent with previous measurements, which found $H^+(H_2O)_3$[55 u] and $H^+(H_2O)_4$[73 u] to be dominant at 70 km and $NO^+$[30 u] and $O_2^+$[32 u] to dominate at 105 km. Large positive ions have not been detected, neither at 70 km nor at 105 km above our measurement threshold. From our large ion measurements at 70 km the following conclusions may be drawn: The presence of large negative ions suggests that at 70 km, electron attachment to neutral MSPs was sufficiently fast and neutralisation of negative MSPs by photo detachment and recombination with positive ions was sufficiently slow to allow a substantial fraction of MSPs to be negatively charged. The absence of positive MSPs suggests that, neutralisation of positive MSPs by collisions with free electrons was faster than positive MSP formation by uptake of positive ions.

*Data availability.*  The data will be made available through HALO database: https://halo-db.pa.op.dlr.de

*Author contributions.* :
J. Stude: investigation, methodology, formal analysis, project administration, software, visualization, writing – original draft;
H. Aufmhoff: investigation, validation, software, writing – review & editing;
H. Schlager: supervision, writing – review & editing;
M. Rapp: funding acquisition, supervision, writing – review & editing;
F. Arnold: conceptualization, validation, supervision, writing – review & editing;
B. Strelnikov: conceptualization, writing – review & editing

*Competing interests.* The authors declare that they have no conflict of interest.

*Acknowledgements.* The authors wish to express their thanks to the mobile rocket base (MORABA) and Andøya Space Center (ASC) for their intense support and help. We also like to thank, Bernhard Preissler[†], Robert Lindemann and Systemhaus Technik for their dedicated technical support. We further acknowledge discussions at working groups 414 and 437, at the International Space Science Institute (ISSI) in Berne, Switzerland. Project PMWE was funded by the German Federal Ministry for Economic Affairs and Energy and by the German Aerospace Center (DLR) under grant 50OE1402.

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
