# Peer review of "A novel rocket borne ion mass spectrometer with large mass range: instrument description and first flight results"

_Atmospheric Measurement Techniques, 2020_

## Referee Comment (RC1) · Anonymous Referee #1 · 28 Jul 2020

This manuscript proves the description of the ROMARA instrument, which is a cryocooled quadrupole mass analyzer. ROMARA is an updated version of a successful instrument flown a few decades ago. The manuscript presents the basic operation principle and characteristics of the instrument, along with preliminary data. This article will be a good introduction for the more detailed analysis of the data and their interpretation and thus its publication is recommended. With that said, I do have some constructive feedback that I encourage the authors to consider.

- Line 30/31. Particles with masses of 'tens...of atomic mass units' are called MSP particles. I would think that these masses would fall under atomic or molecular ions,

instead of MSPs.

- Fig. 1. I find it strange to talk about the density of particles with masses down to 10 u.

- Line 107. Perhaps it would be better talking about fractions (e.g. percentages) of ions passing through the quadrupole, instead of 'few ions'.

- Line 134. The unit of data rate would be kbits/sec. Either call it data volume, or provide the actually rate.

- Line 144. m/z 5 – 2075. The = sign is missing.

- Line 160. It would be useful to provide some key information about the conditions for the launch. For example, the Sun elevation angle, or the orientation of the payload wrt to the Sun. Later in the manuscript scattered UV photons are mentioned.

- Line 172. Maybe I have missed it, but was there a numerical analysis that considered the effect of the angle of attack on the transmission of ions through the quadrupole filter? This would be useful to discuss to some extent.

- Fig. 6&7 and the text describing them present the data in the units of count rates. It would be useful to provide an estimate how to such rates convert to number density.

- As a general comment, I have missed some level of discussion of how the CEM detection probability varies with the mass of the ions. Is there any information on this? Apologies if it is there and I have missed it.

- Lines 208 and 244: It appears that the instrument measures significantly more negatively charged ions/particles than positively charged ones. This is a potentially significant issue that in my opinion needs to be treated carefully. In particular, the quasineutrality of the plasma is not discussed. What would possibly be the cations or positive charge carriers that remain undetected? I am not sure if I can agree with the statement starting on line 244 that the neutralization of positive MSPs due to free electrons is a viable mechanism. The large number of negative particles already suggest that the electrons are scavenged from plasma. Several models have been published on the charge balance of MSPs that could provide some guidelines on how to interpret the observation for the given condition (solar elevation angle, for example). It is probably a good idea to briefly mention or discuss these models, just to provide a background for reader. If there is a significant disagreement between the models and the data, it should be stated.

- Another general comment: I am not sure if I have seen a discussion how heavy neutral MSP particles could possibly affect the measurements. Such particles may pass through the Q/m filter unaffected and be detected. Any information of this that is worth discussing? My guess is that at higher altitude and the corresponding higher angles of attack this become less of an issue, but perhaps at the lowest  ${\sim}2$  degree angle they have a direct path to the detector from the orifice.

СЗ

---

## Referee Comment (RC2) · Anonymous Referee #2 · 11 Aug 2020

This paper describes what seems to be a very exciting instrument for detecting heavy charged ions in the Earth's mesosphere and lower thermosphere. It is essentially a resurrected mass spectrometer which is now run in a mode for rapid scanning up to 2000 amu, including an estimate of the numbers of even heavier ions, and measures in both positive and negative ion modes. The detector is a channel electron multiplier coated with MgO to improve sensitivity to heavy ions. Results from the inaugural flight in northern Norway are reported.

The instrument has been designed to detect meteoric smoke particles, which are the nm-sized particles which form from the condensation of the metallic vapors produced

by meteoric ablation. One of the goals of the flight was to determine the possible role of these particles in radar echoes that are seen during polar winter in the mesosphere. The science that has come out of the measurements is largely left for future papers, and this paper focuses on the instrumental description and performance.

Promising results are obtained. These seem to be in accord with previous measurements and models which have shown that smoke particles should mostly be negatively charged below about 90 km. My only substantive question is around charge balance. The positive ion spectrum at 69 km in Figure 6 shows a much lower integrated count rate than the negative ion spectrum at 70 km in Figure 7. Do these counts needs to be scaled in some way to compare them directly? The total positive ions should essentially equal the total negative ions, since electrons should almost all be attached to molecules or particles at this height, even during the day.

The paper is well written and appropriately illustrated. Some minor points for correction are listed below. One final point: the data availability needs to be specified – location and electronic address.

Minor points:

line 5: major

line 20: there are some more recent references (from the Leeds group) which might be appropriate here:

Frankland, V. L.; James, A. D.; Feng. W.; Plane, J. M. C. (2015): The uptake of HNO3 on meteoric smoke analogues, Journal of Atmospheric and Solar-Terrestrial Physics, 127, 150-160, 127, 150-160.

James A. D., J. S. A. Brooke, T. P. Mangan, T. F. Whale, J. M. C. Plane, and B. J. Murray (2018), Nucleation of nitric acid hydrates in polar stratospheric clouds by meteoric material, Atmospheric Chemistry and Physics, 18, 4519-4531.

line 59: presumably the payload detaches from the rocket motor at some point before

measurements commence? This should be made clear here, since it sounds as though the motor is attached throughout flight.

line 59 and elsewhere: supersonic is usually written as a single word

line 65: the end of this sentence reads a little strangely. Perhaps rewrite to something like ". . . cryopump also provides structural support."

line 67: "on the ground. . ."

line 68: ". . .transmission, independent bias potentials can be applied to the intake cone, lens and quadrupole"

Figure 2 caption: what is "Bat." – presumably battery, but this should be spelled out.

line 111: "..). For the inline. . ."

line 118: ". . .the ram direction"

line 120: ". . .of a standard . . ."

line 121: ". . .atmosphere with a composition. . ."

line 170: ". . .operate up to 49"

line 172: "Thus ions enter the instrument"

line 173: "In constrast to 70 km,"

line 182: "In constrast to the calibration"

---

## Author Comment (AC1) · 15 Oct 2020

Dear Referee #1 Thank you very much for your helpful review. Answers and actions to your comments are provided below.

"Line 30/31. Particles with masses of 'tens: : :of atomic mass units' are called MSP particles. I would think that these masses would fall under atomic or molecular ions, instead of MSPs"

CHANGED: hundreds to millions of atomic mass units [u]

We refer to Huntens bin sizes starting at 0.2 nm radius. With the different density

[Figure]

assumptions, this leads to a minimum of 40 u which is probably more a "molecular ion" than a MSP.

"Fig. 1. I find it strange to talk about the density of particles with masses down to 10u."

CHANGED: Figure reprinted to fit 0.2 to 10 nm. The line from Bacher et al. was adjusted to his actual measurements (start from 3022 u)

"Line 107. Perhaps it would be better talking about fractions (e.g. percentages) of ions passing through the quadrupole, instead of 'few ions'."

CHANGED: ... only small fractions of ions above 10e4 ...

"Line 134. The unit of data rate would be kbits/sec. Either call it data volume, or provide the actually rate."

CHANGED: ...the data volume per spectrum is about 10 kbyte."

"Line 144. m/z 5 – 2075. The = sign is missing."

CHANGED: ...peak height was determined with about 17.5 ( 5 u peak width for Kr) and the mass range from m/z 5 to m/z 2075.

"Line 160. It would be useful to provide some key information about the conditions for the launch. For example, the Sun elevation angle, or the orientation of the payload wrt to the Sun. Later in the manuscript scattered UV photons are mentioned."

ADDED: At the given time and location a solar zenith angle of 61.6° is calculated, the direction of the launch was 330° azimuth.

"Line 172. Maybe I have missed it, but was there a numerical analysis that considered the effect of the angle of attack on the transmission of ions through the quadrupole filter? This would be useful to discuss to some extent."

The simulations carried out should only show that the RF-only mode is not simply a high-pass mode. In general the increasing AoA reduces the amount of incident

ions (see eq. 3). More refined trajectory simulations would require more detailed assumptions on the incident particles properties and a higher geometric detail. Albeit this is very interesting, it would go beyond the scope of the paper.

"Fig. 6&7 and the text describing them present the data in the units of count rates. It would be useful to provide an estimate how to such rates convert to number density."

ADDED: additional section 3.5 Charge balance at 70 km altitude

In principle these count rates can be converted to ion densities (N=c/(A v). c as counts, A as intake area, v rocket speed. This leads to ion densities of $N_+$ = 32 cm-3 and $N_-$ =379 cm-3, however some factors are not considered (see below).

"As a general comment, I have missed some level of discussion of how the CEM detection probability varies with the mass of the ions. Is there any information on this? Apologies if it is there and I have missed it."

ADDED: additional section 3.5 Charge balance at 70 km altitude

The working principle of a CEM detector is the generation of secondary electrons at the cone of the CEM. This mainly depends on the incident particle (mass, speed, angle) and the material of the cone (secondary electron yield). There are numerous publications on detection efficiency for atomic ions like noble gases, hydrogen and oxygen but little or none for heavy molecules. Thus for example C. A. Keller, and B. H. Cooper for positive and negative oxygen report 0.6 to 0.7 at 2 keV (ROMARA has 1.8 keV). Or Krems et al reporting similar values (2 keV) for oxygen 0.75 and 0.15 for xenon, reaching 1 with sufficient post acceleration. For micro channel plates, that use the same electron multiplying principle, Gilmore reported efficiencies of about 0.02 for 2352 u at 2 keV. For electrons a MgO coated CEM was tested by Manalio et al. to be about 3 times more efficient than uncoated.

"Lines 208 and 244: It appears that the instrument measures significantly more negatively charged ions/particles than positively charged ones. This is a potentially significant issue that in my opinion needs to be treated carefully. In particular, the quasineutrality of the plasma is not discussed. What would possibly be the cations or positive charge carriers that remain undetected?"

ADDED: additional section 3.5 Charge balance at 70 km altitude

For 69/70 km: Quasineutrality requires: $N_+ = N_- + N_e$. Measured electron densities: SAURA: $\sim$ 500 e/cm3 (Latteck 2019). With: c+ $\sim$26 kHz and c- $\sim$300 kHz (200 kHz light ions + 100 kHz heavy ions): $N_+$ = 32 cm-3 and $N_-$ =379 cm-3. Payload charging to positive values in the sunlight could explain this discrepancy, as the positivly charged rocket would attract negativly charged particles and repel positivly charged.

"I am not sure if I can agree with the statement starting on line 244 that the neutralization of positive MSPs due to free electrons is a viable mechanism. The large number of negative particles already suggest that the electrons are scavenged from plasma."

The electron density is measured to be about 500 e/cm3 (Latteck 2019) and thus electrons would be availabe for neutralization of positive MSPs.

"Several models have been published on the charge balance of MSPs that could provide some guidelines on how to interpret the observation for the given condition (solar elevation angle, for example). It is probably a good idea to briefly mention or discuss these models, just to provide a background for reader. If there is a significant disagreement between the models and the data, it should be stated."

In this instrument paper we wanted to focus on the instrument and first results. We are working on a follow-on paper, where we compare our measurements to the Sodankylä Ion Chemistry Model (SIC) See: Verronen et al. 2005, doi:10.1029/2004JA010932

"Another general comment: I am not sure if I have seen a discussion how heavy neutral MSP particles could possibly affect the measurements. Such particles may pass through the Q/m filter unaffected and be detected. Any information of this that is worth discussing? My guess is that at higher altitude and the corresponding higher angles of

attack this become less of an issue, but perhaps at the lowest ∼2 degree angle they have a direct path to the detector from the orifice."

A neutral particle requires an angle of attack below 0.85° (angle between center and apertures) to pass the intake orifice and the exit aperture of the quadrupol. Heavy particles have a low angular spread at mesospheric temperatures. Thus the probability to enter is low for AoA>1°. Further, any neutral particle effects both, positive and negative ion measurements. Thus a neutral particle signal should be present during positive and negative ion mode. However, in negative ion mode secondary electrons from neutral particles might be detected. To briefly test that, we used UV LEDs to stimulate the CEM, independent on the applied voltages for the different ion modes. We measured a 3 times higher count rate in negative ion mode as in positive ion mode. As the photons generate secondary electrons in front of the CEM cone, these electrons are more easily captured in negative ion mode. However, we do not measure a heavy ion signature in positive ion mode that is in the order of 3 lower than in negative ion mode.

---

## Author Comment (AC2) · 15 Oct 2020

Dear Referee #2 Thank you very much for your helpful review. Actions and answers to your comments are provided below.

"My only substantive question is around charge balance. The positive ion spectrum at 69 km in Figure 6 shows a much lower integrated countrate than the negative ion spectrum at 70 km in Figure 7. Do these counts needs to be scaled in some way to compare them directly?"

ADDED: additional section 3.5 Charge balance at 70 km altitude

[Figure]

In principle these count rates can be converted to ion densities (N=c/(A v). c as counts, A as intake area, v rocket speed. For 69/70 km: Quasineutrality requires: N+ = N- + Ne. Measured electron densities: SAURA: $\sim$ 500 e/cm3 (Latteck 2019). With: c+ $\sim$26 kHz and c- $\sim$300 kHz (200 kHz light ions + 100 kHz heavy ions): N+ = 32 cm-3 and N- =379 cm-3 (see chapter 3.5 for more details). Payload charging to positive values in the sunlight could explain this discrepancy, as the positivly charged rocket would attract negativly charged particles and repel positivly charged.

"One final point: the data availability needs to be specified – location and electronic address."

CHANGED: The data will be made available through HALO database: https://halo-db.pa.op.dlr.de

Minor points:

"line 5: major"

CORRECTED

"line 20: there are some more recent references (from the Leeds group) which might be appropriate here: Frankland, V. L.; James, A. D.; Feng. W.; Plane, J. M. C. (2015): The uptake of HNO3 on meteoric smoke analogues, Journal of Atmospheric and Solar-Terrestrial Physics, 127, 150-160, 127, 150-160. James A. D., J. S. A. Brooke, T. P. Mangan, T. F. Whale, J. M. C. Plane, and B. J. Murray (2018), Nucleation of nitric acid hydrates in polar stratospheric clouds by meteoric material, Atmospheric Chemistry and Physics, 18, 4519-4531."

ADDED: references

"line 59: presumably the payload detaches from the rocket motor at some point before measurements commence? This should be made clear here, since it sounds as though the motor is attached throughout flight."

CHANGED: ...supersonic speed of the payload...

"line 59 and elsewhere: supersonic is usually written as a single word"

CORRECTED

"line 65: the end of this sentence reads a little strangely. Perhaps rewrite to something like ": : : cryopump also provides structural support.""

CHANGED: ... which serves also as structural support...

"line 67: on the ground"

CORRECTED

"line 68: ... transmission, independent bias potentials can be applied to the intake cone, lens and quadrupole ...

CORRECTED

"Figure 2 caption: what is "Bat." – presumably battery, but this should be spelled out."

CHANGED

line 111: "..). For the inline: : :"

CHANGED: For the line mode...

line 118: ": : :the ram direction"

CORRECTED

line 120: ": : :of a standard : : :"

CORRECTED

line 121: ": : :atmosphere with a composition: : :"

CORRECTED

line 170: ": : :operate up to 49"

CORRECTED

line 172: "Thus ions enter the instrument"

CORRECTED

line 173: "In contrast to 70 km,"

CORRECTED

line 182: "In contrast to the calibration"

CORRECTED

---

## Author Response (AR2)

Revised Submission
Associate Editor Decision: Publish subject to minor revisions (review by editor) (10 Nov 2020) by Troy Thornberry
Comments to the Author:
Dear Dr. Stude and co-authors,

I would like to commend you on your efforts to address the comments and critiques of the reviewers to the submitted manuscript. I think the revised manuscript is in good condition, but would ask the authors to consider the following comments.

The last sentence in section 2.2 ('Heavier particles...') would benefit from somewhat fuller description.
*Particles with much larger mass such as MSPs, will be less affected by the shock formed by air molecules (Hedin et al., 2007; Asmus et al., 2017) and thus penetrate it more efficiently.*

It would perhaps be better to include the operational specifications of the mass spectrometer (the mass spectral scan rate/time, the number of channels) in the electronics section (2.3) along with the speed of the rocket since this provides information on the vertical resolution of the measurements.
*We moved the corresponding information to the electronics section and the vertical resolution is now given in the figure caption.*

In section 2.4 referring to the lower panel of Fig. 5, is there information about instrument performance that can be drawn from the steeper falloff with increasing mass of the measured signal to that of the theoretical?
*The effect causing the signal to fall off is the residual ions from the system from leaking air and hydrocarbon contaminations. The text was changed accordingly and a reference given to the spectrum in figure 6A. See below the data from the calibration with the residual signal included.*

[Figure]

*Figure 1: calibration data with residual gas spectrum*

The abstract contains information about the launch that does not appear in the text--please include full details in section 3.1.
I did not see that you addressed one reviewer's question about payload separation.
*We expanded the description in the section launch conditions with the following details:*
*After separating the nose cone at 52.2 km the instrument cap was jettisoned at 54.5 km and ROMARA measurements began. At 60.0 km the motor was separated from the payload. The payload hit the echo at its decaying tail before reaching an apogee of 121.4 km.*

The additional text addressing the reviewers' questions about charge balance is a good addition, but the description of the potential impact of payload charging could be better introduced or more fully developed.

The section focuses on the charge balance at ~70 km, but the difference between the spectra at ~55 km is even starker (essentially no positive ions).

*We changed the section name and added 55 km into the description.*

*We further expanded the payload potential section to address the problem more appropriately.*

This list contains changes made to the document "AMT_20201119_markedup.pdf".

- Line 53: added references: Strelnikov 2020 / Staszak 2020
- Line 131: rephrased
- Line 140: added text
- Line 145: rephrased
- Line 155: added text
- Line 156: rephrased, added text
- Line 158: rephrased, deleted text
- Line 171: rephrased, added text
- Line 196: updated number
- Line 201: updated number
- Figure 6: added text, updated numbers in figure
- Line 203: updated number
- Line 209: updated number
- Figure 7: added text, updated numbers in figure, deleted times in 7B
- Line 218: updated number
- Line 221: rephrased
- Line 224: updated number
- Line 226: added text
- Line 229: rephrased
- Line 231: added text
- Line 238: updated number
- Line 241: rephrased
- Line 247: added text
- Line 249: added text
- Line 252: added text
- Line 257 - 263: rephrased
- Line 264 – 296: added text

[revised manuscript text omitted]